# Application of Polyaniline for Flexible Semiconductors

Ana-Maria Mocioiu [1], Ioan Albert Tudor [1] and Oana Cătălina Mocioiu [2],*

1   National Research & Development Institute for Non-Ferrous and Rare Metals-IMNR, 102 Biruinţei Blvd, Pantelimon, 077145 Ilfov, Romania; ammocioiu@imnr.ro (A.-M.M.); atudor@imnr.ro (I.A.T.)
2   Ilie Murgulescu Institute of Physical Chemistry of the Romanian Academy, 202 Splaiul Independenţei, 060021 Bucharest, Romania
*   Correspondence: omocioiu@icf.ro

**Abstract:** "In situ" polymerization method was used to develop PANI-PSSA /textile. Polyaniline doped with polystyrene sulfonic acid (PANI-PSSA) used as coatings for textiles were obtained by aqueous and emulsion route. The emulsion route uses chloroform as solvent. Polymerization has been achieved in one step on the wool or polyamide textiles. For coated and uncoated textiles, dried at room temperature, were characterized structurally by Infrared Spectroscopy with Attenuated Total Reflectance (ATR), morphologically by Scanning Electron Microscopy (SEM), and by Atomic Force Microscopy (AFM) and electrically. The synthesis methods lead to differences in structure, morphology and properties of the coated polyamide and wool textiles.

**Keywords:** one step polymerization; aqueous route; emulsion route; polyamide textile; wool

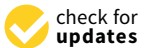



## 1. Introduction

Flexible materials gain interest and have a wide variety of applications. The required properties of flexible materials vary according to the applications. The structure and the properties of textiles, as thickness, lightweight, flexibility, deformability, breathability, recommend them as substrates [1,2]. Sensors can be deposited on the textile substrates using conductive polymers. Conducting polymers, such as polyaniline (PANI), are used in numerous applications in sensors, in textiles with antimicrobial properties, or in smart textiles for data transfer, electrotherapy, defense technology and electromagnetic materials for monitoring health [1–7]. For many of the applications mentioned above polymeric textiles must have electrical properties. Polyaniline can have different conductivity values in a range of 15 orders of magnitude. The conduction of polyaniline is achieved through a mechanism that includes the exchange of both electrons and protons [4,5]. Also, ionic conductivity is influenced by the porosity of the material and is closely linked to the mobility of ions inside the system [5].

Polyaniline-polystyrene blends were obtained previously as membranes and thin films deposited on solid substrates.

Cho used emulsion synthesis with the chloroform to achieve PANI–PSS–Pt composites with spread evenly of platinum nanoparticles [8]. In interfacial synthesis, polymerization of polyaniline has been progressed by diffusion of aniline monomer from chloroform layer to aqueous solution layer [8].

Polystyrene sulfonate (PSS) acts as the bridge between the inter-chain of polyaniline (PANI) to form a continuous spatial network when water is used. This spatial network structure of PANI-PSS was observed by Huang in case of films deposited on Pt substrate [9]. Polystyrene sulfonate is used as the proton exchange polymer. Besides increasing the proton conductivity, PANI- PSS can form a spatial network structure [9].

The conductive blends of nanostructured polyaniline-clay composite (PANICN) and Polystyrene (PS) for electromagnetic interference shielding, were developed from by a one-step matrix-assisted emulsion polymerization of aniline salt of 3-pentadecyl phenol-4-

sulphonic acid (3-PDPSA) in clay. A product based on renewable resources is 3-PDPSA, a liquid derived from cashew shell [10]. The product is obtained at low costs [10].

Huang reported the doping of polyelectrolyte, PSS subsequently into PANI structure, through a doping–dedoping–redoping route. Incorporation of PSS into PANI resulted from advantageous parameters for electrochromic applications. The modification observed in morphology through SEM is consistent with the branched structure for polyaniline doped with polystyrene sulfonate. The branched structure resulted have good thermal stability, evident in TGA data [11].

Li obtained composites deposited on a glass substrate for practical applications in polymer electronics [12]. Polyaniline-poly (sodium 4-styrene sulfonate) particles prepared from weak acids or low concentration of strong acids can form stable colloids in water or buffers over a range of pH from 1 to 11 [12]. The acids used in the study were acetic acid, propionic acid, hydrochloric acid, methyl sulfonic acid, nitric acid, sulfuric acid, and DL-camphor sulfonic acid (98%). The size of composite particles during the polymerization can be controlled by the number of negative charges on PSS [12].

In other application, the film of polyaniline nanowires has been deposited on the stainless steel electrode through galvanostatic polymerization of the aniline and utilized for the fabrication of the catalyst electrode. Polyaniline nanowire was doped with styrene sulfonic acid by sequential doping–dedoping–redoping process, to result a PANI (NW)-PSS spatial matrix. The obtained nanowires were deposited on a platinum electrode. The electrode PANI (NW)-PSS/ Pt with a spatial distribution of the particles showed a better catalytic performance besides oxidation of methanol than the PANI (NW)/Pt [13].

Gold nanoparticles has been assembled on the surface of polystyrene and polyaniline core–shell nanocomposite (PS@PANI). An electrochemical sensor for the immobilization of HL-60 leukemia cells was fabricated. This biosensor was low-cost and disposable, which implied that the composite has the potential for clinical applications [14].

The membranes [15] were prepared by mixing high impact polystyrene with a conducting polyaniline. Three dopants, dodecilbenzene sulfonic acid (DBSA), p-toluene sulfonic acid (p-TSA), and camphor sulfonic acid (CSA), were used. These membranes have been obtained by mechanical mixture in screw extruder and solvent dissolution [15]. The lower resistance and better transport properties of the membranes processed with solvent allow them to be more used than those processed by mechanical mixture [15].

Haberko reported possible to transpose a pattern of hydrophobic SAM deposited on a gold surface into a thin film composed of polyaniline doped with camphorsulfonic acid and polystyrene using the spin-coating method [16]. Hierarchical structures are formed, with the micron scale variations corresponding to substrate pattern periodicity, and the submicron (∼0.3 m) features corresponding to polyaniline columns [16].

Polyaniline/polystyrene blends were prepared by polymerization with molar ratio of 1:2 and PSS sulfonated at the degrees of 13%, 21%, and 30 mol% by Rubinger [17]. The conductivity of the blends increases with the sulfonation degree. The doping level of the polyaniline increases when the amount of $SO_3H$ groups in the polymeric acid increases. The smoothness of the blend surface increased with the sulfonation degree. The values of parameters obtained from resistivity fitting suggest that 3D inter-chain conduction occurs preferably than 1D conduction along the polymeric chain [17].

In general polyaniline colloids are synthesized using polymeric templates specially polyacrylic acid, polyvinyl pyridine and polystyrene sulfonate [18–21].

Electrically conductive core–shell nanoparticles (PANI/PSPSS) has been prepared by coating poly (styrene co-styrene sulfonate) (PSPSS) nanoparticles with polyaniline (PANI) [22]. PSPSS core particles has been obtained in the micro-emulsion system in a nitrogen atmosphere and coated with PANI by in situ polymerization at low temperature [22]. The highest conductivity of PANI/PSPSS pellets is 1.7 S/cm [22].

The performance of the organic light emitting diodes (OLED) increased by chemical oxidation polymerization of a water-dispersed polyaniline-polystyrene sulfonate (PANIPSS) prepared in the presence of excess HCl and used as a HIL [23]. Molar ratio was

PANI:PSS = 1:12 [23]. Compared with PEDOT-PSS (5.18 eV), the higher work function of PANI-PSS (5.27 eV) enhanced the hole injection from the ITO to HTL. The properties of the PANI-PSS film coated on the ITO presented a reduced surface roughness, higher work function, and good transparency, compared to poly(3,4-ethylene dioxy- thiophene) (PEDOT)-PSS [23].

Shavandi reported the deposition of the polystyrene on the wool textile [24]. Styrene needs to be dissolved in methanol in order to be successfully deposited onto wool with little homopolymerization [24]. This technique is not applicable to other monomers, in particular to acrylates, because of the high amount of copolymer formation occurs and prevents the successful wool coating [24]. Incorporation of styrene with ethyl acrylate lead to a successful coating with a minimum formation of unwanted homopolymer [24].

Wool fabric with chitosan/polyaniline was obtained by in-situ polymerization method [25]. Oxygen plasma pre-treatment was used to etch the scale layer of wool and to introduce active free radicals which can form hydrogen bonds with the –OH of the chitosan. The treatment improved the binding force among chitosan, polyaniline and wool and it was implied in obtaining of uniform conductive layer [25].

A molecular template was used to integrate of a conductive polymer (polyaniline) into wool-based textiles [26]. The efficiency of the polymerization/coating process is increased by the template that localizes the reaction within the textile [26]. The presence of the molecular template leads to the formation of an adherent, uniform and stable layer with conductive properties [26]. The templated polyaniline- wool Nylon Lycra fabrics were suitable for application such as wearable strain gauge materials that can be used for biomechanical monitoring [26].

In this paper new conducting textiles (wool, polyamide), coated with polyaniline (PANI)–polystyrene sulfonic acid (PSSA), were obtained. Two synthesis methods were used: water based and emulsion in chloroform. Coated textiles by aqueous route and emulsion route were obtained and a comparison between the two routes was made.

Literature describes methods for obtaining coatings based on polyaniline with other polymers/monomers or metals (Au, Pt, etc.), with higher production costs [13,14,25]. However, based on our knowledge, the polyaniline-polystyrene based textiles prepared by emulsion route using chloroform, as well as PANI-PSSA coated textiles, were not reported until now.

## 2. Materials and Methods

### 2.1. Reagents and Chemicals

For the preparation of conductive textiles, the following materials were used: aniline 99% (Sigma-Aldrich, Saint Louis, MI, USA), polystyrene sulfonate acid (PSSA, M = 75,000) (Sigma-Aldrich, Saint Louis, MI, USA), ammonium persulfate (Sigma-Aldrich, Saint Louis, MI, USA), distilled water, chloroform (Sigma-Aldrich, Saint Louis, MI, USA).

Textiles from the Romanian market were used in this research. Polyamide fabric has a white color with a thickness of 0.25 mm and a mass of 112 $g/m^2$. Wool textile has the thickness of 0.9 mm and a mass of 184 $g/m^2$.

### 2.2. Synthesis Procedures

#### 2.2.1. Aqueous Route Synthesis

In a reactor equipped with a stirrer, 8 mL polystyrene sulfonate acid (PSSA, *M* = 75,000) and 1 g of aniline in 160 mL distilled water was added. The mixture was stirred for 10 min, after that the textile was introduced. Above the reaction mass was poured a solution of 2.5 g ammonium persulfate in 40 mL distilled water, dripped for 15 min at a temperature between 0–10 °C. The mixture was stirred for 90 min after dripping. The textile was removed from the reaction mass, was washed with distilled water and dried at room temperature. Textiles used were polyamide textile and wool textile. Ratio textile:liquid was 1:7. The molar ratio of aniline:oxidant was 1:1.3. The coated polyamide textile obtained by aqueous route was labeled P-Aq-PANI-PSSA and the wool was labeled W-Aq-PANI-PSSA.

### 2.2.2. Emulsion Route Synthesis

In a reactor equipped with stirrer, 8 mL polystyrene sulfonate acid (PSSA, $M$ = 75,000) in 250 mL chloroform and 1 g aniline in 150 mL chloroform were introduced. The mixture was stirred for 10 min, and the textile was immersed. Above the reaction mass was poured a solution of 2.5 g ammonium persulfate in 40 mL distilled water, dripped for 15 min at a temperature between 0–10 °C. The emulsion obtained was stirred for 90 min after the dripping. The textile was removed from the reaction mass, was washed with distilled water and dried at room temperature. Textiles used were polyamide textile and wool textile. Ratio textile:liquid was 1:12. The molar ratio aniline:oxidant was 1:1.3. The coated polyamide textile obtained by emulsion route was noted P-E-PANI-PSSA and the wool textile was noted W-E-PANI-PSSA. In case of the emulsion route, chloroform improved solvent evaporation at room temperature and maintains temperature below 10 °C in the reaction batch.

### 2.3. Characterization of the Coated Textiles

Fourier transformed infrared spectra of textiles were obtained at room temperature using a Cary 630 spectrometer. The measurement of the total attenuated reflection (ATR) was performed by mounting the sample on a Diamond crystal. Absorption spectra were obtained by summing of 32 scans in the range of 650–1800 cm$^{-1}$ and 2300–4000 cm$^{-1}$, with a resolution of 4 cm$^{-1}$. All the important bands of textiles coated with doped polyaniline are situated in those intervals of wavenumbers. Organic and inorganic groups were identified. Surface morphology of textiles was observed using scanning electron microscope (SEM) Quanta 250 -FEI model (FEI, Hillsborough, OR, USA). Atomic Force Microscopy (AFM) analysis of uncoated and PANI-PSSA coated textiles were performed using a Nanovea microscope, model Nanosurf Nanite B (Nanosurf AG Liestal, Liestal, Switzerland), 110 microns. The measurements were performed in air at room temperature using contact mode and static force operation mode. The surface resistivity of the coated textiles was measured according to standard SR EN 1149-1:2006 [27] employing the 2 electrodes method, using a PROSTAT 800 m (PROSTAT Corporation, Bensenville, IL, USA). The conductivity was established by the calculus formula. Electrochemical Impedance Spectroscopy (EIS) was performed to investigate semiconductor properties of coated textiles and to evaluate their electronic and ionic conductivities. The measurements in a range of 100 kHz to 0.01 Hz using 3 mV AC perturbation, were made using a SOLARTRON 1260 A (AMETEK Scientific Instruments, Thames, UK).

### 3. Results and Discussion

#### 3.1. Scanning Electron Microscopy

The surface morphology of coated materials was determined by electron microscopy using a Quanta 250, FEI apparatus (FEI Company). Figure 1a–d shows the SEM images of textiles coated with conductive polymer PANI-PSSA. As one can see in figures both wool and polyamide textiles were coated by uniform and continuous thin films. The smooth aspect of coatings was determined by polystyrene sulfonate acid doping. It must be underlined that thick films were obtained in the emulsion while in the aqueous reaction were obtained thin films.

From the above discussion, it is conceivable that the surface morphology of PANI is influenced by the incorporation of PSSA into PANI structure from aqueous solution or emulsion. The SEM images (Figure 1) exhibited the samples obtained by emulsion route have grains with a porous structure, while Aq-PANI-PSSA shows a better cohesion with compact layer structure. The change in morphology of PANI-PSSA arises from the effect of the solvent on the orientation of polymer on substrate. The PANI-PSSA polymer that was obtained by emulsion route has the morphology of grains compacted on the fiber. The morphologies of polyaniline coatings are different and they look different from the other textile coatings obtained with HCl as dopant [3].

a) W-Aq-PANI-PSSA (wool)          c) W-E-PANI-PSSA (wool)

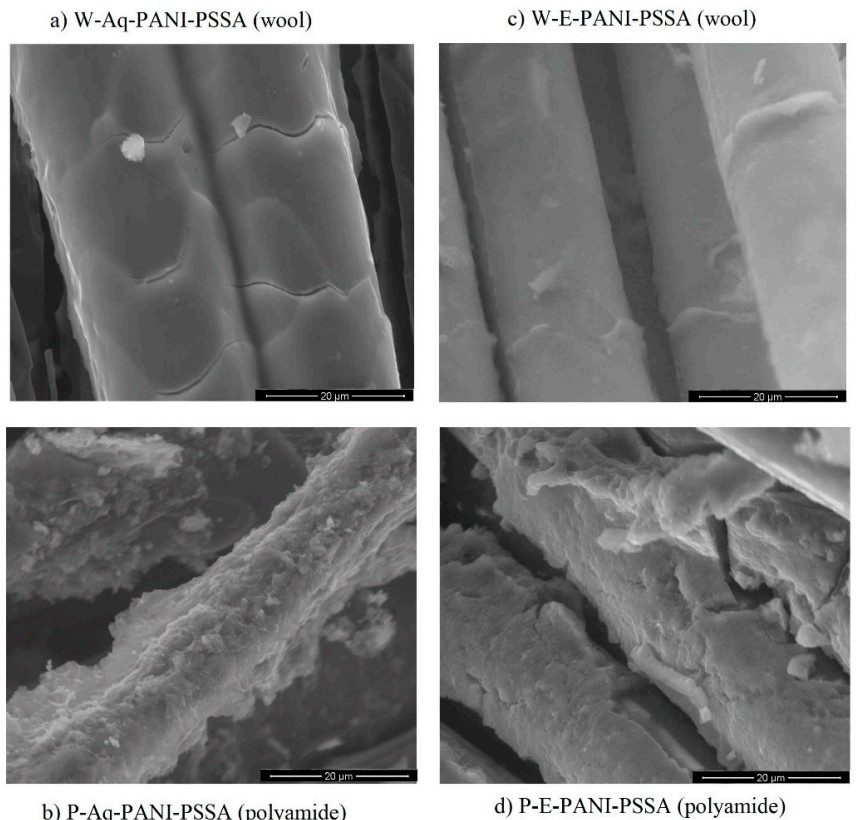

b) P-Aq-PANI-PSSA (polyamide)     d) P-E-PANI-PSSA (polyamide)

**Figure 1.** SEM images of coated textiles: (**a**) W-Aq-PANI-PSSA (wool); (**b**) P-Aq-PANI-PSSA (polyamide); (**c**) W-E-PANI-PSSA (wool); (**d**) P-E-PANI-PSSA (polyamide).

### 3.2. Atomic Force Microscopy

The 2D and 3D AFM images of the uncoated polyamide textile surface is presented in Figure 2a. The AFM images of P-E-PANI-PSSA (polyamide) is presented in Figure 2b and show uniform coating. The AFM images in Figure 2c show that the coated wool had periodic stripes on its surface probably induced by orientation of the molecular chains along the fiber axis.

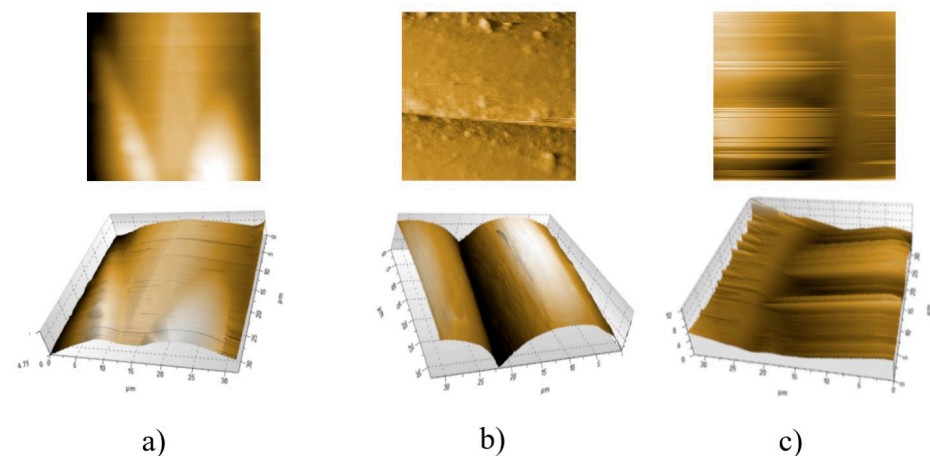

a)                    b)                    c)

**Figure 2.** 2D and 3D AFM images: (**a**) polyamide; (**b**) P-E-PANI-PSSA (polyamide); (**c**) W-E-PANI-PSSA (wool).

### 3.3. Infrared Spectroscopy

Figure 3 presents ATR spectra of the textiles obtained by coating of polyamide and

wool with doped polyaniline. The band frequencies and their assignations are given in Table 1.

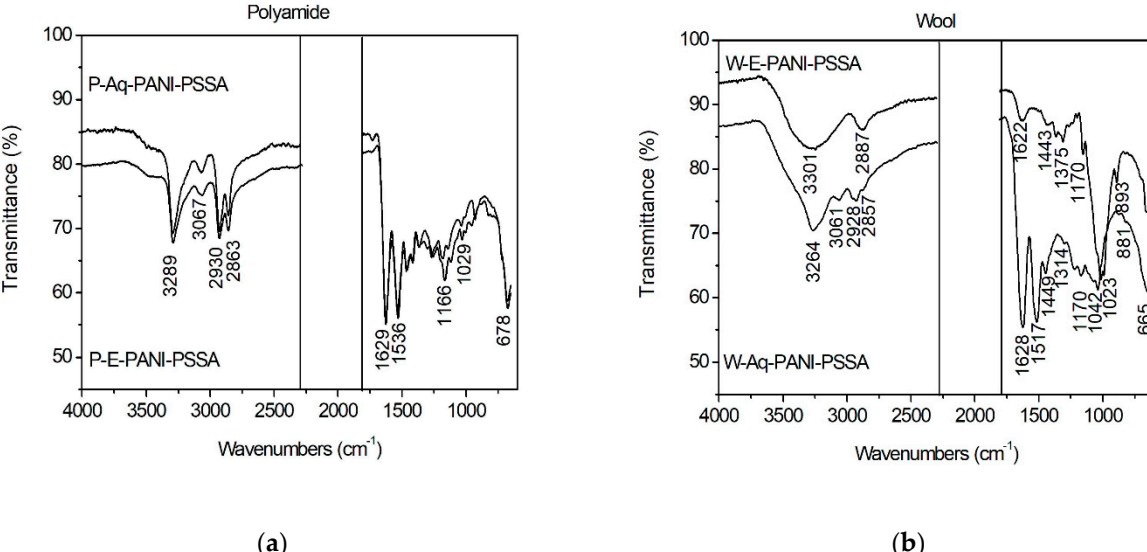

**Figure 3.** ATR spectra of PANI-PSSA coated textiles: (**a**) polyamide and (**b**) wool.

**Table 1.** The frequencies characteristic to coated textiles and their assignments.

| Assignment | W-Aq-PANI-PSSA | P-Aq-PANI-PSSA | W-E-PANI-PSSA | P-E-PANI-PSSA |
|---|---|---|---|---|
| C–N, N–H stretching | 3264, 3061 | 3295, 3073 | 3301 | 3289, 3067 |
| C–H in benzene | 2928 | 2931 | - | 2930 |
| C–H stretching | 2857 | 2857 | 2887 | 2863 |
| C=O in amide | - | 1733 | - | 1733 |
| N=Q=N | 1628 | 1629 | 1622 | 1629 |
| N–B–N | 1517 | 1536 | 1536 | 1536 |
| - | 1449 | 1467 | 1443 | 1467 |
| - | - | 1412 | - | 1412 |
| C–N in Q–B states | 1314, 1172 | 1370, 1184 | 1375, 1313, 1159 | 1370, 1301, 1166 |
| - | - | 1264 | 1270 | 1264 |
| - | 1227 | - | 1233, 1202 | - |
| C–H in plan bending | - | 1147 | - | 1122 |
| $SO_3^-$ group of dopant | 1042 | 1042 | 1023 | 1029 |
| $CH_3$ group attached by phenyl ring | - | 931, 1004 | 992 | 955, 1004 |
| Para di-substituted aromatic rings | 881 | 795 | 893 | 795, 838 |
| C–H deformation out-of –plan | 665 | 678 | 665 | 678 |

Q = quinoid form, B = benzene ring.

The comparison of the spectra in the case of polyaniline on polyamide obtained by aqueous route (P-Aq-PANI-PSSA) and in emulsion (P-E-PANI-PSSA) shows similar bands characteristic of PANI. The intensities of the bands at 1628 and 1517 cm$^{-1}$ can be assigned to polyaniline with 50% quinoid form and 50% benzene ring. In the spectrum of wool, the band position and the intensity is different. In the spectrum of W-E-PANI-PSSA the band of quinoid form at 1629 cm$^{-1}$ has higher intensity than those of the benzene ring at 1536 cm$^{-1}$. In the W-Aq-PANI-PSSA spectrum the bands of quinoid form and benzene ring at 1629 and 1536 cm$^{-1}$ are characteristic to PANI with equal percent of both forms. The band at 1029 cm$^{-1}$ assigned to $SO_3^-$ group of dopant has higher intensity in the W-E-PANI-PSSA spectrum than in the spectrum of W-Aq-PANI-PSSA. This behavior can be explained by a better polymerization on wool when the aqueous synthesis route is

used, due to hydrophobicity of wool. The bands at 795–893 cm$^{-1}$ can be assigned to para di-substituted aromatic rings indicating polymer formation.

### 3.4. Electrical Properties

Since most synthetic fibers are hydrophobic, they have low moisture retention and have a much higher electrical resistance on the surface. In comparison, natural fibers have high moisture retention and exhibit higher conductivity on the surface, which allows the accumulated charges to dissipate in the surroundings [28].

The resistivity of materials was measured in agreement with standard SR EN 1149-1:2006. The conductivity of obtained materials was calculated from resistivity with the following formula:

$$\sigma = 1/\rho, \tag{1}$$

where σ is conductivity and ρ is resistivity.

In Table 2 are presented the results of measured resistivity according to standard SR EN 1149-1:2006 [27] and calculated conductivity of coated obtained materials.

**Table 2.** Resistivity measured according to Romanian standard SR EN 1149-1:2006 [27] and calculated conductivity of coated obtained materials.

| Coated Textiles | Experimental Surface Resistivity (Ω·cm) SR EN 1149-1:2006 | Calculated Surface Conductivity (S/cm) | Aspect/Colour |
|---|---|---|---|
| W-E-PANI-PSSA | $7.3 \times 10^3$ | $1.4 \times 10^{-4}$ | Uniform coated, dark green |
| P-E-PANI-PSSA | $8.8 \times 10^3$ | $1.1 \times 10^{-4}$ | Uniform coated, dark green |
| W-Aq-PANI-PSSA | $1 \times 10^5$ | $1 \times 10^{-5}$ | Uniform coated, green |
| P-Aq-PANI-PSSA | $7.4 \times 10^4$ | $1.4 \times 10^{-5}$ | Uniform coated, dark green |

The experimental surface resistivity values measured for uncoated textiles were $1.2 \times 10^{11}$ Ω·cm for polyamide and $8.6 \times 10^9$ Ω·cm for wool. The coated textiles show improved properties and experimental surface resistivity measured between $10^3$ and $10^5$ Ω·cm.

The conclusion of data from Table 2 is that the coated textiles obtained by emulsion route exhibit higher conductivity than the coated textiles obtained by aqueous route. There is a small difference between the conductivity values of coated textiles obtained by emulsion, but the polyamide has a better conductivity. When aqueous route was used for coating, the wool textile shows a better conductivity. Results obtained in this work are comparable with data reported in the literature, such as electrical conductivity $10^{-5}$ S/cm in SBA-15/PANI nanocomposites [29]. The textiles with PEDOT: PSS: graphene were reported as flexible and wearable e-textiles for smart clothing in general and electrocardiogram (ECG) biomedical applications in particular with electrical conductivity of $10^5$–$10^2$ Ω·cm [29]. They used aqueous route modified for syntheses [29].

Impedance spectroscopy technique was used to analyze the electrical characteristics which are important for a practical application. Nyquist plots of the PANI-PSSA coated textiles are shown in Figure 4. It is known that Nyquist profile is dedicated to the electrochemical interactions on the surfaces of electrodes, in this case coated textile.

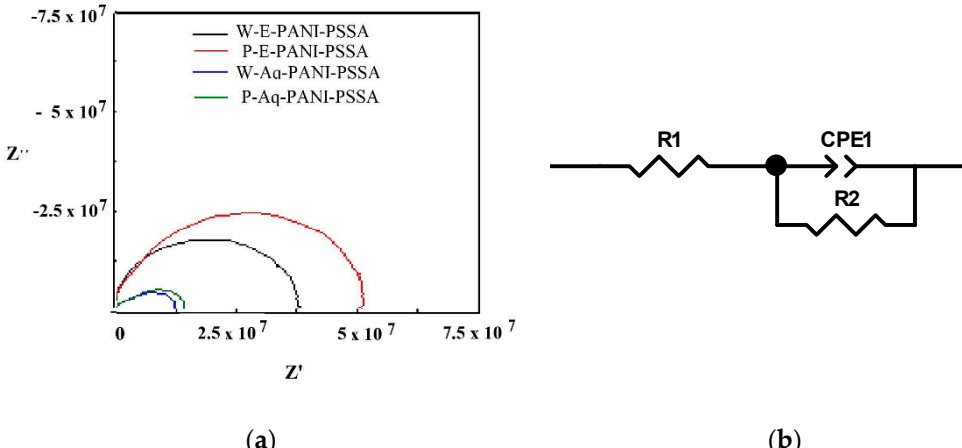

**Figure 4.** (**a**) Nyquist plots of the coated textiles and. (**b**) Model of circuit fitted based on Nyquist plot for materials obtained by emulsion route.

Complex impedance spectra of textiles coated in emulsion were fitted by using an equivalent circuit that includes different contributions to the overall impedance: electrons and ions in the PANI-PSSA coating and electrode contact. In Table 3 are presented results of the model of circuit elements for materials obtained by emulsion route.

**Table 3.** Circuit elements and errors for coated textiles obtained by emulsion route.

| W-E-PANI-PSSA Element | Value | Error | Error% |
|---|---|---|---|
| R1 | $2.0412 \times 10^6$ | $2.0023 \times 10^5$ | 9.80 |
| CPE-T | $2.028 \times 10^{-12}$ | $8.0468 \times 10^{-14}$ | 3.96 |
| CPE-P | 0.99 | 0.0048 | 0.48 |
| R2 | $4.9819 \times 10^7$ | $2.9389 \times 10^5$ | 0.58 |
| **P-E-PANI-PSSA Element** | **Value** | **Error** | **Error%** |
| R1 | 39468 | 8093 | 20.50 |
| CPE-T | $3.727 \times 10^{-12}$ | $8.4457 \times 10^{-14}$ | 2.21 |
| CPE-P | 1.009 | 0.0021 | 0.21 |
| R2 | $3.6972 \times 10^7$ | $1.8207 \times 10^5$ | 0.49 |

In our investigation, the impedance spectra show one semicircle at room temperature. The semicircle shape shows the semiconductor properties as is explained in the EIS apparatus handbook. It is well known that the semiconductor is a material whose resistivity is between that of conductors and insulators [30]. An electric field can change the resistivity of semiconductors [30]. In a metal conductor, the current is represented by the flow of electrons. As example, copper, the best conductor material, has an electrical resistivity of $1.7 \times 10^{-8}$ $\Omega \cdot$m ($1.7 \times 10^{-6}$ $\Omega \cdot$cm) [30]. Some materials such as glass ($10^{11}$ to $10^{15}$ $\Omega \cdot$m) and sulfur ($10^{15}$ $\Omega \cdot$m), which have high resistivity, are very good electrical insulators [30]. Semiconductors are weak conductors because a current is needed to move electrons. In a semiconductor the current is represented either by the flow of electrons or by the flow of "gaps" in the electronic structure of the material [30]. Typically, the diameter of the arc within the high frequency region reflects the charge transfer resistance, which is related to interfacial processes of counter-ions at the electrode/electrolyte interface. W-Aq-PANI-PSSA has the smallest value of arc ($1.7 \times 10^7$) among the measured coated textiles, which is in agreement with the conductivity from Table 2. The low frequency region of the

spectra is associated with adsorption processes, microscopic charge transfer, and surface roughness. The slope of the curves in the low frequency region is assigned to the pseudo capacitive properties. The synthesis with chloroform instead of water leads to obtaining coated textiles with semiconductor properties necessary for application as a sensor.

## 4. Conclusions

Polyaniline-polystyrene coated textiles obtained can be used as flexible semiconductors. Two synthesis routes were used, as well as polymerization in one step directly on textiles. The molar ratio of aniline: oxidant was 1:1.3 in both syntheses. The morphology of doped polyaniline film deposited on the textiles is highly influenced by the synthesis route.

Both synthesis routes improve conductive properties of coated textiles by $10^6$–$10^8$ times compared to the initial textiles. Another conclusion to be drawn from results is that the synthesis on the emulsion route is most suitable for conductive polyamide textiles and the aqueous synthesis is better for conductive wool textiles.

The electrochemical properties were evidenced by Nyquist plots. Semiconductors are weak conductors because a current is needed to move electrons. The synthesis with chloroform instead of water leads to obtaining coated textiles with semiconductor properties necessary for application as a sensor.

**Author Contributions:** A.-M.M. and O.C.M. made conceptualization, formal analysis, investigation, writing—original draft, and I.A.T. participate at formal analysis (AFM). All authors have read and agreed to the published version of the manuscript.

**Funding:** This research was partially funded by Ministry of Education and Research of Romania, by national project "Core Program"; project number 19190501.

**Institutional Review Board Statement:** Not applicable.

**Informed Consent Statement:** Not applicable.

**Data Availability Statement:** Data available in a publicly accessible repository.

**Acknowledgments:** This work was supported in part by the Core Program; project number 19190501; MEC Romania.

**Conflicts of Interest:** The authors declare no conflict of interest.

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
