# Peer review of "Application of Polyaniline for Flexible Semiconductors"

_coatings, doi:10.3390/coatings11010049_

Round 1

Reviewer 1 Report

Oana Cătălina Mocioiu et al. display a new methods to construct flexible conductivity textiles via polyaniline-polystyrene on emulsion route. It is interesting and important for material community.  I recommend it acceptance after minor revised. 

  1. For the characterization of morphology, AFM or TEM should be done for detail analysis of surface of related film, especially for comparing the polyamide and wool
  2. For the convenience, all the main peaks should be marked in the spectrum. 

Author Response

Thank you for the comments. We improved the article. AFM measurements were added in the article and the colleague who made the measurements in the list of authors. we passed the main bands in the FTIR spectra.

Reviewer 2 Report

Unfortunately, the subject of the publication is mediocre. The preparation of PANI-PSSA polymer is widely described in the literature. After reading it, I didn't notice any particular differences from the use of chloroform. The shape of the impedance indicates that there is practically no diffusion control at low frequencies. The layer resistance is very high. I believe that the publication is not suitable for publication.

Author Response

Thank you for the comments. We improved the article.

Reviewer 3 Report

  1. The quality of the paper is not as much as an academic paper. The authors are asked to check the paper with more results and better discussion. In the paper, just the results have brought and no complete discussion.
  2. English should be revised or the paper should re-written by a native.
  3. Introduction is so short with many references for a small part of text. For example, for the first 3 lines, 9 references is reported. It is better to extend/explain better and bring the most related articles for each section. If one reader want to know more about every small paragraph, s/he is referred to lots of references and it is so confusing. In line 29, very total sentence with [1-16] references! Duty of author is to choose the best references if the text is repeated in many.
  4. Novelty of work is not clear or not well written. In Introduction, the author supposed to tell what a problem is and what their solution is. In the other words, what is missing in the other papers that this paper analyzed? What new knowledge sb can get with reading this article, which is not included in the other publications?
  5. In normal academic writing, the authors bring review of the other papers at first, and then, refer to their work in the last paragraph.
  6. In section" Reagents and chemicals", the text should be rewritten better. Also, characteristics of the textiles were used like, thickness, g/m2, …
  7. "a" and "b" are not specified on the figures. For example, in Fig. 1, it should be written clearly which part is "a" and which part is "b".
  8. Figure 1 has very low quality; it should be replaced with the higher one with more quality.
  9. Caption of tables should be rewritten. Table 3 has no caption.
  10. "ool" in the conclusion, means wool?

Author Response

Thank you for the comments.  We have improved the article and we have included in this version all the corrections mentioned by you. Best regards.

Round 2

Reviewer 2 Report

The improvement of the manuscript did not affect the still low novelty and the poor quality of the results. The presentation of the results and the impedance matching are incorrect.

Reviewer 3 Report

Fine